# Real-world effects of alcohol on heart rate, sleep, and physical activity by age and sex

**Gregory J. Grosicki**[1]*, **Austin T. Robinson**[2], **Michael J. Joyner**[3,4], **Jason R. Carter**[5], **William von Hippel**[1,6], **David M. Presby**[7,8,9], **Finnbarr Fielding**[1], **Jeremy A. Bigalke**[5], **Jeongeun Kim**[1], **Christopher Chapman**[1], **Kristen E. Holmes**[1]

**1** Department of Performance Science, WHOOP, Inc., Boston, Massachusetts, United States of America, **2** Department of Kinesiology, Indiana University, Bloomington, Indiana, United States of America, **3** Department of Anesthesiology and Perioperative Medicine, Mayo Clinic, Rochester, Minnesota, United States of America, **4** Department of Physiology and Biomedical Engineering, Mayo Clinic, Rochester, Minnesota, United States of America, **5** Department of Health, Human Performance, and Recreation, Baylor University, Waco, Texas, United States of America, **6** Research with Impact, Brisbane, Queensland, Australia, **7** Department of Research, Algorithms, and Data Science, WHOOP, Inc., Boston, Massachusetts, United States of America, **8** Swiss Institute of Bioinformatics, Lausanne, Switzerland, **9** Department of Computational Biology, University of Lausanne, Lausanne, Switzerland

* greg.grosicki@whoop.com

## Abstract

Alcohol consumption acutely disrupts physiology and behavior. Yet, the modifying effects of age, biological sex, and health behaviors are not well understood. In this retrospective cohort study, we analyzed 5,109,185 person-days from 20,968 participants and fit generalized additive models to estimate within-person associations between alcohol intake and nocturnal resting heart rate (RHR), heart rate variability (HRV), sleep duration, and next-day physical activity. Models were stratified by age and sex, and adjusted for drinking frequency, body mass index, weekday/weekend, and season, and accounted for between-person differences via person-mean centering. We also assessed whether drinking earlier in the day, longer post-drinking sleep, and reducing physical activity attenuated disruptions. Acute alcohol consumption was associated with dose-dependent increases in nocturnal RHR and reductions in HRV, alongside decreases in sleep duration and next-day physical activity. These changes were more pronounced in females than males and in younger than older adults: consuming one drink more than personal average, compared with one less, was associated with an increase in RHR by 2.8 bpm (99.9% CI: 2.7, 2.9) in females and 2.4 bpm (99.9% CI: 2.3, 2.4) in males, while HRV declined by 3.8 ms (99.9% CI: -4.1, -3.5) in females and 3.3 ms (99.9% CI: -3.5, -3.1) in males. Drinking earlier in the day, obtaining longer post-drinking sleep, and reducing activity each reduced these effects. Alcohol consumption acutely disrupts cardiovascular regulation, sleep duration, and next-day physical activity, with stronger disruptions in females and younger adults. Behavioral modifications may mitigate these disruptions.

which permits unrestricted use, distribution, and reproduction in any medium, provided the original author and source are credited.

**Data availability statement:** The data and code that support the findings of this study are not publicly available due to intellectual property concerns of WHOOP, Inc. Deidentified participant data and a data dictionary may be made available upon reasonable request to WHOOP (via research@whoop.com). Access will require submission of a methodologically sound proposal, approved by WHOOP's research team, and a signed data use agreement. The timeframe for response to requests will be ≥4 weeks. The preregistered statistical analysis plan is archived at Open Science Framework (https://osf.io/zf5sp).

**Funding:** This work as supported by WHOOP, Inc. through salary support provided to authors GJG, WvH, DMP, FF, JK, CC, and KEH. No specific grant numbers are associated with this work. WHOOP, Inc. (https://whoop.com/) provided general institutional support but had no role in study design, data collection and analysis, decision to publish, or preparation of the manuscript.

**Competing interests:** I have read the journal's policy and the authors of this manuscript have the following competing interests: The study was funded by WHOOP, Inc. GJG, WvH, DMP, FF, JK, CC, and KEH are employees of WHOOP, Inc., and GG, DMP, FF, JK, CC, and KEH hold stock options as part of their employee compensation, which represents a potential financial interest. These authors contributed to the study design, data curation, statistical analysis, and manuscript preparation. All analyses followed a prespecified workflow, and analytic decisions were made solely within the research and data science teams. Marketing, enterprise, and other non-research groups at WHOOP, Inc. had no role in study approval, data interpretation, manuscript preparation, or decisions regarding submission or publication. The remaining authors declare no competing interests.

## Author summary

We set out to understand how a night of drinking changes heart rate, sleep, and next-day physical activity in everyday life, and whether these changes might differ by age and biological sex. Using data from nearly 21,000 adults who wore a wearable sensor, we compared each person's nights with alcohol to their own nights without alcohol. After drinking, resting heart rate during sleep was higher, heart rate variability was lower, people slept less, and they were less active the next day. These dose-related changes were larger in females than males, and larger in younger than older adults. We also looked for simple habits that might lessen these effects. Drinking earlier in the day, getting more sleep after drinking, and keeping exercise lighter on drinking days were each linked to smaller adverse effects. Our findings offer population-level evidence that even low volumes of drinking can affect nightly recovery and next-day physical activity, and offer practical guidance to lessen some of the adverse effects of drinking for people who choose to drink.

## Introduction

Alcohol is one of the most widely consumed drugs worldwide [1]. Contrary to earlier research, growing consensus now indicates that even low to moderate alcohol consumption poses measurable health risks [2,3]. Despite increased awareness of alcohol's long-term health impacts [2], there is a critical knowledge gap regarding its acute effects in real-world settings. Advances in wearable technology enable the measurement of real-time physiological and behavioral responses to alcohol in ecologically valid contexts. In doing so, these tools offer a scalable way to assess how alcohol consumption influences cardiovascular physiology, sleep, and physical activity in everyday life. Leveraging a large, real-world dataset, we examined within-person associations between alcohol intake and nocturnal resting heart rate (RHR), heart rate variability (HRV), sleep duration, and next-day physical activity, and whether these associations vary by age, biological sex, and are moderated by health behaviors. We focused on RHR and HRV as key established indicators of autonomic and cardiovascular function [4,5], and on sleep duration and physical activity as leading health behaviors that shape these metrics and overall health [6,7].

   Most insights into alcohol's acute physiological repercussions come from controlled laboratory studies, which demonstrate marked disruptions of sleep and cardiovascular regulation [8,9]. While valuable, these studies are often constrained by small, homogenous samples and artificial conditions. One of the few sizable real-world studies to date reported that alcohol's cardiac autonomic impact was similar between males and females, but more pronounced in younger compared to older adults [10]. However, that study relied on a smaller sample (n = 4,098; 12,411 days) than the present study, required only one day with and without alcohol, and focused solely on the first three hours of sleep. To our knowledge, no prior

investigation has examined cardiovascular, sleep, and physical activity responses to alcohol consumption in a dataset as large or diverse as the present study. Importantly, we also assessed several modifiable health behaviors including timing of alcohol consumption, sleep duration after drinking, and physical activity on the day of drinking. Our aim was to provide actionable insights to inform individual strategies and public health efforts to reduce alcohol-related health impacts.

## Results

### Participant characteristics

The participant flow diagram is presented in Fig 1. Following eligibility filtering, the analytic sample included 20,968 participants (10,025 female and 10,943 male), contributing 5,109,185 person-days of observation. Table 1 presents alcohol use patterns and key cardiovascular and health behavioral metrics by age and biological sex.

Males consumed alcohol more frequently, and in greater quantities, than females across all age groups. Drink timing relative to sleep was similar between sexes. Males exhibited lower RHR than females in each age group, and higher HRV values than females in the younger but not older age groups. Females consistently slept longer than males, while activity differences emerged only in the 60 + yr age group where males were more active.

Older adults drank more frequently but consumed fewer drinks per occasion. Drink-to-sleep timing did not vary significantly by age. RHR increased and HRV decreased progressively with age. Sleep duration steadily declined with advancing age. Physical activity decreased through midlife, but peaked in the 60 + yr cohort.

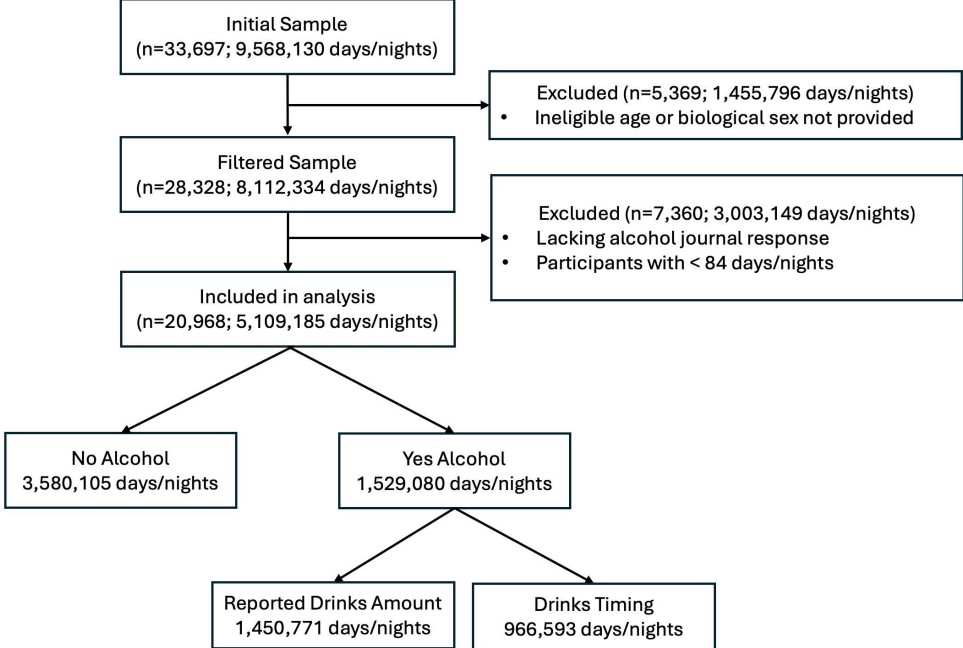

**Fig 1. Participant flow diagram depicting inclusion and exclusion criteria for the study sample.** Of the initial 33,697 participants contributing 9,568,130 days/nights, 5,639 were excluded due to missing or ineligible demographic data. An additional 7,360 participants were excluded due to missing alcohol journal responses or contributing fewer than 84 days/nights of data. The final sample contained 20,968 participants and 5,109,185 days/nights of data. Alcohol was not consumed on 3,580,105 days/nights. On 1,529,080 days/nights with alcohol intake, 1,450,771 included a reported number of drinks and 966,593 included a reported time of last drink. For drinking days lacking a reported drinking quantity (5.1% of total), the individual's person-level median drink amount was imputed.

**Table 1. Alcohol, cardiovascular, sleep, and physical activity outcomes by age and sex.**

| | Drinking Frequency (%) | Drinks on Drinking Nights (#) | Average Drinks Per Day (#) | Drink-to Bedtime (min) | Resting Heart Rate (bpm) | Heart Rate Variability (ms) | Sleep Duration (hrs) | Physical Activity (AU) |
|---|---|---|---|---|---|---|---|---|
| **Females (n = 10,025)** | | | | | | | | |
| 20-29 yrs (n = 2275) | 15.1 (21.0)[b,c,d,e,^] | 2.8 ± 2.2[b,c,d,e,^] | 0.38 (0.61)[c,d,e,^] | 213 (78)[c] | 60.7 ± 7.6[c,d,e,^] | 69.1 ± 28.2[b,c,d,e,^] | 7.27 (0.79)[c,d,e,^] | 67.1 (77.9)[b,c,e] |
| 30-39 yrs (n = 2308) | 18.5 (27.5)[a,c,d,e,^] | 2.5 ± 1.8[a,d,e,^] | 0.40 (0.69)[c,d,^] | 208 (85)[c,^] | 61.8 ± 7.6[c,d,e,^] | 54.9 ± 23.4[a,c,d,e,^] | 7.23 (0.80)[d,e,^] | 60.3 (67.7)[a,d,e] |
| 40-49 yrs (n = 2334) | 21.8 (34.8)[a,b,d,e,^] | 2.5 ± 1.7[a,d,e,^] | 0.42 (0.82)[a,b,^] | 201 (97) [a,b,e] | 62.5 ± 7.9[a,b,^] | 43.6 ± 18.0[a,b,d,e] | 7.19 (0.83)[a,d,e,^] | 62.3 (67.8)[a,d,e] |
| 50-59 yrs (n = 2151) | 26.4 (40.3)[a,b,c,e,^] | 2.2 ± 1.4[a,b,c,e,^] | 0.48 (0.86)[a,b,^] | 204 (92)[^] | 62.4 ± 7.8[a,b,^] | 38.2 ± 16.2[a,b,c] | 7.06 (0.91)[a,b,c,^] | 68.8 (75.1)[b,c,e] |
| 60 + yrs (n = 957) | 32.0 (46.4)[a,b,c,d,^] | 1.9 ± 1.1[a,b,c,d,^] | 0.47 (0.85)[a,^] | 214 (94)[c,^] | 62.7 ± 7.5[a,b,^] | 36.3 ± 15.9[a,b,c,^] | 7.01 (0.99)[a,b,c,^] | 72.2 (105.8)[a,b,c,d,^] |
| **Males (n = 10,943)** | | | | | | | | |
| 20-29 yrs (n = 2052) | 16.7 (24.3)[b,c,d,e,^] | 4.1 ± 3.4[b,c,d,e^] | 0.59 (1.01)[c,d,e,^] | 205 (84)[c,d] | 56.1 ± 7.0[b,c,d,e,^] | 74.1 ± 27.7[b,c,d,e,^] | 7.02 (0.82)[c,d,e,^] | 64.5 (68.2)[b,e] |
| 30-39 yrs (n = 2212) | 22.3 (32.6)[a,c,d,e,^] | 3.3 ± 2.6[a,c,d,e,^] | 0.64 (1.08)[c,d,e,^] | 200 (81)[d,^] | 57.7 ± 7.3[a,c,d,e,^] | 57.7 ± 23.8[a,c,d,e,^] | 6.93 (0.91)[c,d,e,^] | 55.1 (64.4)[a,d,e] |
| 40-49 yrs (n = 2233) | 28.3 (39.3)[a,b,d,e,^] | 3.1 ± 2.4[a,b,d,e,^] | 0.72 (1.21)[a,b,d,^] | 192 (90)[a,^] | 59.1 ± 7.9[a,b,^] | 44.7 ± 18.6[a,b,d,e] | 6.84 (0.94)[a,b,e,^] | 59.7 (72.7)[d,e] |
| 50-59 yrs (n = 2269) | 35.0 (46.0)[a,b,c,e,^] | 2.9 ± 2.0[a,b,c,e,^] | 0.77 (1.32)[a,b,c,e,^] | 190 (86)[a,b,e,^] | 59.8 ± 7.9[a,b,^] | 37.3 ± 17.0[a,b,c] | 6.76 (0.99)[a,b,e,^] | 71.9 (82.7)[b,c,e] |
| 60 + yrs (n = 2177) | 40.2 (53.3)[a,b,c,d,^] | 2.4 ± 1.6[a,b,c,d,^] | 0.77 (1.28)[a,b,d,^] | 195 (92)[d,^] | 59.0 ± 7.9[a,b,^] | 39.9 ± 23.1[a,b,c,^] | 6.67 (1.17)[a,b,c,d,^] | 83.4 (97.5)[a,b,c,d,^] |

Values averaged from both drinking and non-drinking days and presented as mean ± standard deviation for normally distributed variables and median (interquartile range) for non-normally distributed variables.

a-e indicates statistically significant difference from the specified age group within the same sex, a = 20–29 yrs, b = 30–39 yrs, c = 40–49yrs, d = 50–59yrs, e = 60 + yrs.

^ indicates statistically significant difference between males and females within the same age group.

P < 0.001, adjusted for multiple comparisons.

## Amplified alcohol impact in females compared to males

Within-person associations between alcohol intake and physiological and behavioral outcomes, stratified by biological sex, are illustrated in Fig 2A-2D. Increases in alcohol intake were associated with dose-dependent increases in RHR and decreases in HRV, alongside reductions in sleep duration and lower next-day physical activity (S2 Table). For example, consuming one drink more than usual (vs. one less) was associated with an increase in RHR by 2.8 bpm (99.9% CI: 2.7, 2.9) in females and 2.4 bpm (99.9% CI: 2.3, 2.4) in males, corresponding to moderate standardized effects (ES = 0.61 in females and ES = 0.52 in males). HRV concurrently declined by 3.8 ms (99.9% CI: -4.1, -3.5) in females and 3.3 ms (99.9% CI: -3.5, -3.1) in males, reflecting small effect sizes (ES = 0.30 in females and ES = 0.26 in males). These effects scaled with increasing alcohol intake. Consuming five drinks more than usual (vs. three more) led to HRV reductions of 5.6 ms (99.9% CI: -5.9, -5.3) in females and 5.1 ms (99.9% CI: -5.3, -4.9) in males (ES = 0.45 in females and ES = 0.41 in males). Sleep duration declined progressively with increasing alcohol intake, and next-day activity was likewise lower at higher intake levels, with the largest reductions observed at the highest drinking quantities for both sexes.

Sex-stratified contrasts (S3 Table) indicated that the magnitude of these alcohol-related effects was generally greater in females. Consuming three more drinks than usual was associated with larger increases in RHR (Δ0.54 bpm (99.9%

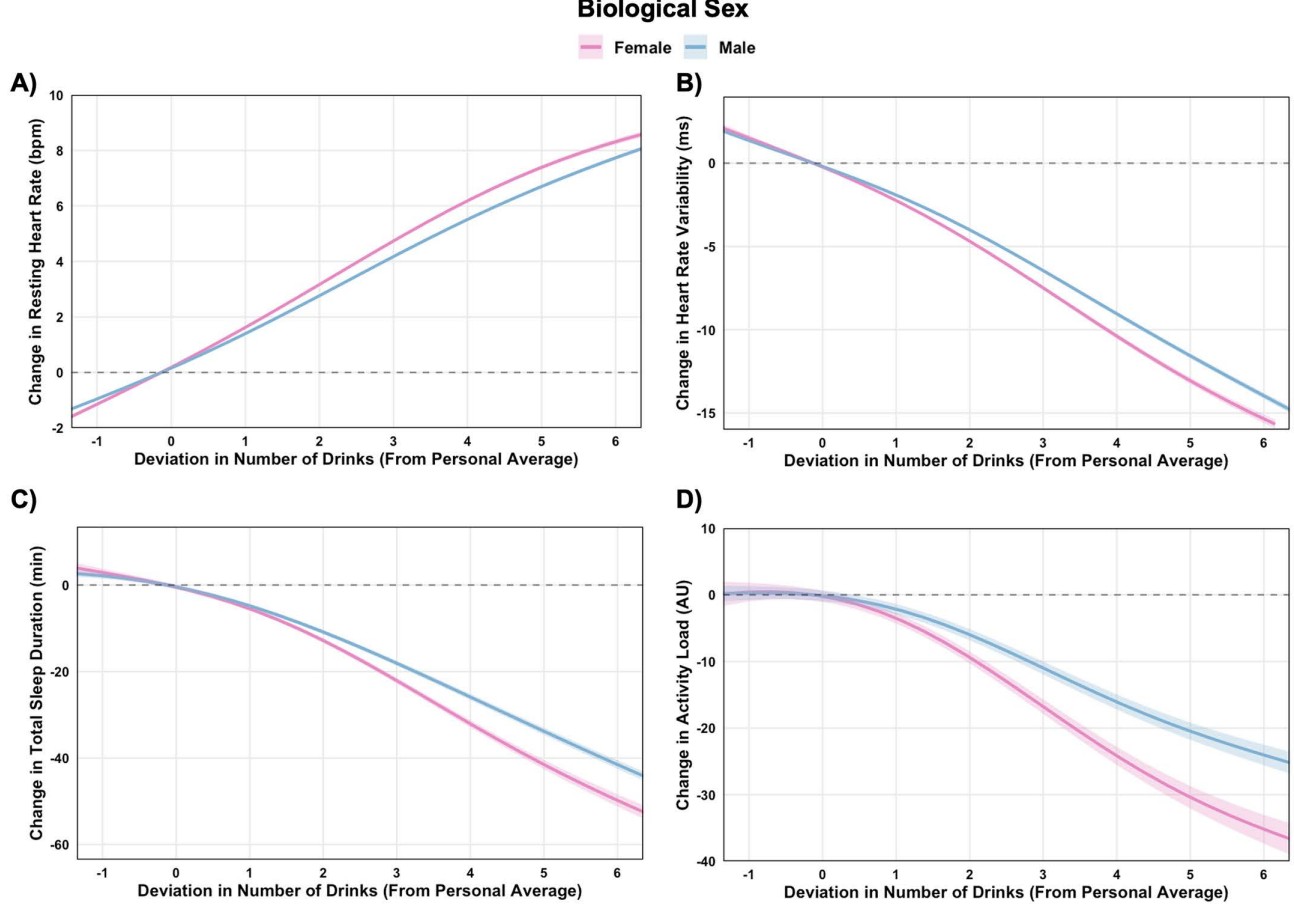

**Fig 2. Within-person associations between deviations in alcohol drink number and physiological and behavioral outcomes, stratified by biological sex.** Generalized additive models estimated changes in resting heart rate (A), heart rate variability (B), sleep duration (C), and next-day activity (D) based on deviations from individuals' personal averages. Dose-response contrasts for drink number are presented in S2 Table, and between-sex comparisons at specific drink quantities are presented in S3 Table.

CI: 0.46, 0.62)) and reductions in HRV (Δ-1.31 ms (99.9% CI: -1.52, -1.10)) in females compared to males (ES = 0.12 for RHR and ES = 0.11 for HRV). Sex differences were also observed for sleep duration and next-day physical activity, both of which tended to decline more in females compared with males. Absolute-value models (S2 Fig) and stratified analyses by baseline outcome (S3 Fig) confirmed these trends.

Earlier drinking attenuated adverse effects on RHR and HRV (Fig 3 A-Fig 3B). For instance, in females drinking 60 min earlier than usual (vs. 60 min later) was associated with a -0.87 bpm reduction (99.9% CI: -0.94, -0.80) in RHR and a 1.5 ms increase (99.9% CI: 1.3, 1.7) in HRV (ES = 0.19 for RHR and ES = 0.12 for HRV; S5 Table). Effects were similar in males (S6 Table). Earlier drinking was associated with slightly shorter sleep in both sexes, with the largest differences at the longest drink-to-bed intervals (Fig 3C). Patterns for next-day activity were more mixed, with many comparisons not significant and a dose pattern that rose and fell (Fig 3D).

## Younger adults exhibit heightened sensitivity to alcohol

Within-person associations between alcohol intake and cardiovascular, sleep, and physical activity across age groups are illustrated in Fig 4A-4D. As with sex-based analyses, increased alcohol consumption was associated with

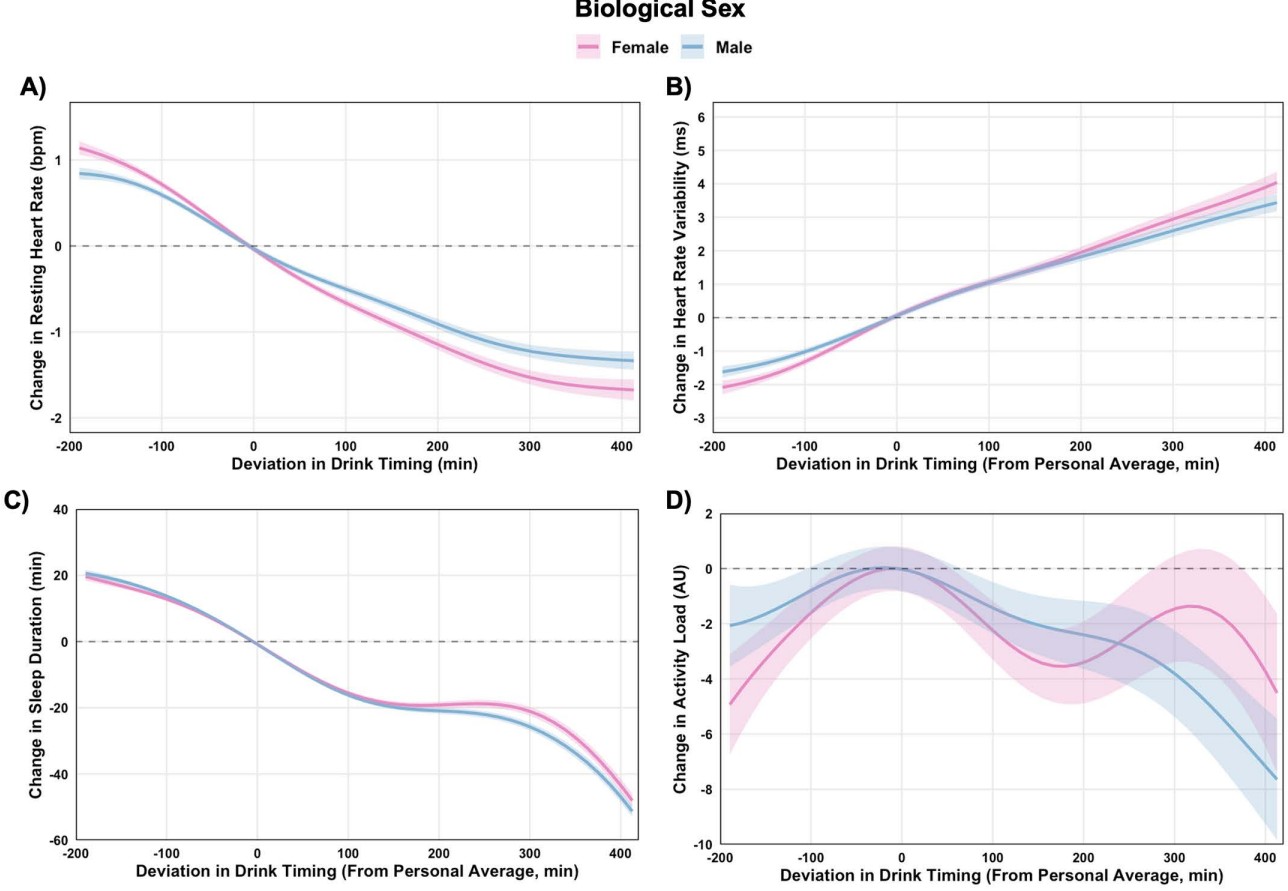

**Fig 3. Within-person associations between deviations in alcohol drink timing and physiological and behavioral outcomes, stratified by biological sex.** Generalized additive models estimated changes in resting heart rate (A), heart rate variability (B), sleep duration (C), and next-day activity (D) based on deviations from individuals' personal averages. Dose-response contrasts for drink timing are provided in S5 Table, with corresponding between-sex comparisons at different timing intervals in S6 Table.

dose-dependent increases in RHR and reductions in HRV, shortening of sleep duration, and lower next-day physical activity (S7 Table). However, an inverse age gradient was observed, whereby disruptions were larger in younger adults (S8 Table). For example, following consumption of five drinks more than usual, HRV declined by 3.0 ms more (99.9% CI: -3.6, -2.5; ES = 0.24) in 20–29 yrs compared to those in their 30's, 3.5 ms more (99.9% CI: -4.1, -2.9; ES = 0.28) in 30–39 yrs compared to those in their 40's, and 2.7 ms more (99.9% CI: -3.4, -2.1; ES = 0.22) in 40–49 yrs compared to those in their 50's. Differences between 50–59 yrs and 60 + yrs were very small and not statistically significant. Similar trends were observed for RHR. Sleep duration exhibited a clear graded reduction with increasing alcohol intake, and next-day activity likewise declined at higher intake levels, but these activity effects were smaller in magnitude. Absolute-value models (S4 Fig) and stratified analyses by baseline outcome (S5 Fig) confirmed these age-associated patterns.

Earlier drinking mitigated adverse effects of drinking on RHR and HRV across all age groups (Fig 5A and 5B). For instance, in 20–29 yr olds shifting alcohol consumption 60 minutes earlier than usual (vs. 60 minutes later) was associated with a 1.2 bpm (99.9% CI: -1.3, -1.1) reduction in RHR and a 3.7 ms (99.9% CI: 3.4, 4.1) increase in HRV (ES = 0.26 for RHR and ES = 0.30 for HRV; S10 Table). Similar patterns were observed in older age groups, though the magnitude

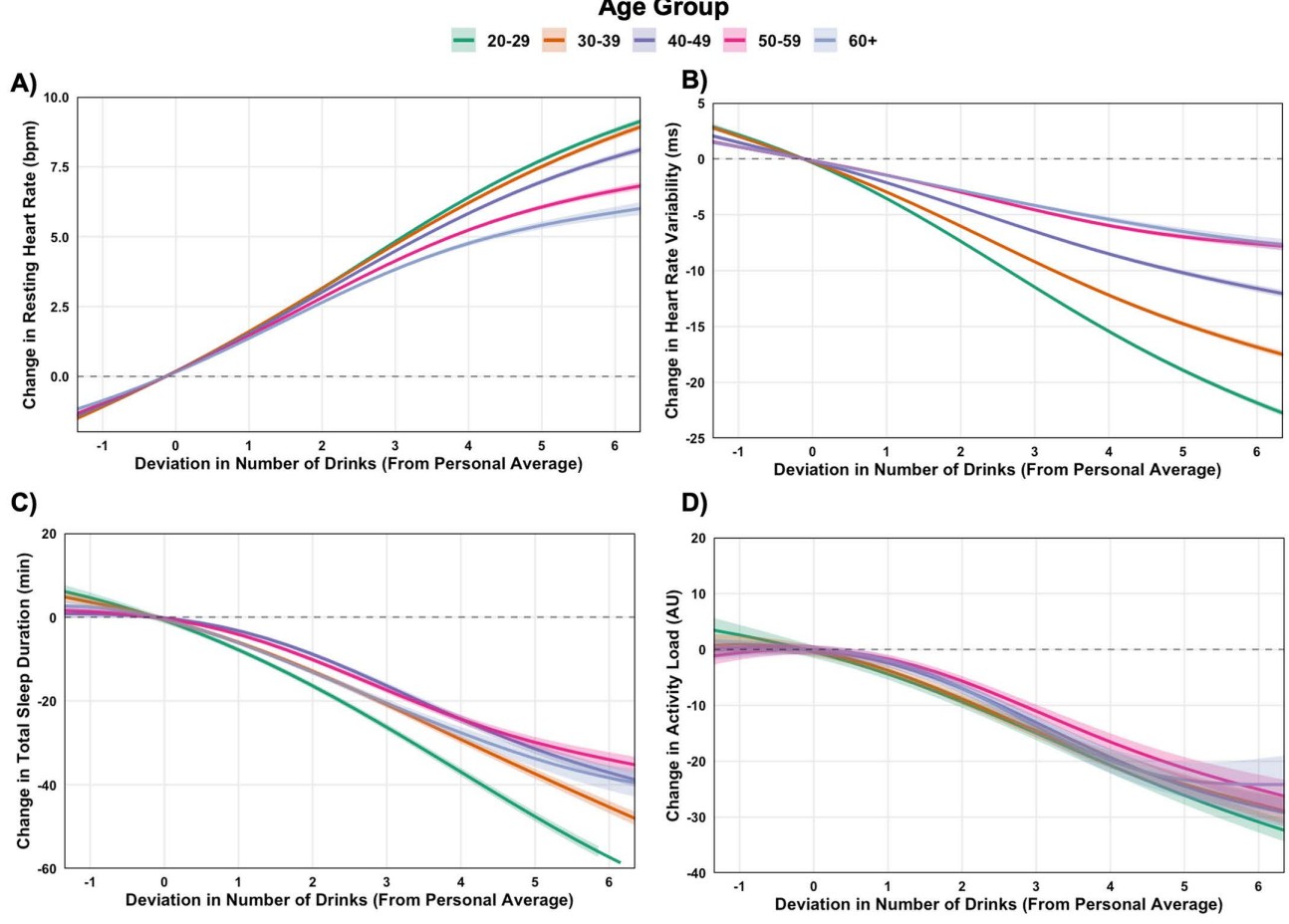

**Fig 4. Within-person associations between deviations in alcohol drink number and physiological and behavioral outcomes, stratified by age group.** Generalized additive models estimated changes in resting heart rate (A), heart rate variability (B), sleep duration (C), and next-day activity (D) based on deviations from individuals' personal averages. Dose-response contrasts for drink number are presented in S7 Table, and between-age group comparisons at specific drink quantities are presented in S8 Table.

of benefits diminished slightly with age (S11 Table). Findings for sleep duration and next-day activity were similar to those from sex-stratified analysis, with earlier drinking associated with slightly shorter sleep (Fig 5C). Activity responses were more mixed, with effects smaller in size and not consistently observed across age groups (Fig 5D).

## Sleep and physical activity moderate the physiological and behavioral impact of alcohol

Fig 6 illustrates within-person associations between alcohol consumption and physiological and behavioral outcomes, stratified by sleep duration (Fig 6A-6C) and day-of-drinking physical activity tertiles (Fig 6D-6F).

Across nearly all alcohol consumption quantities, sleeping more than usual was associated with smaller increases in RHR and attenuated reductions in HRV (Fig 6A-6B). For example, following consumption of five drinks more than usual, RHR was 2.9 bpm (99.9% CI: -3.1, -2.7) lower on high- vs. low-sleep nights, and HRV was increased by 7.2 ms (99.9% CI: 6.8, 7.6) (ES = 0.65 for RHR and ES = 0.58 for HRV; S12 Table). Similarly, next-day activity was 6.0 AU higher (99.9% CI: 2.4, 9.7) on high sleep nights (ES = 0.06; Fig 6C).

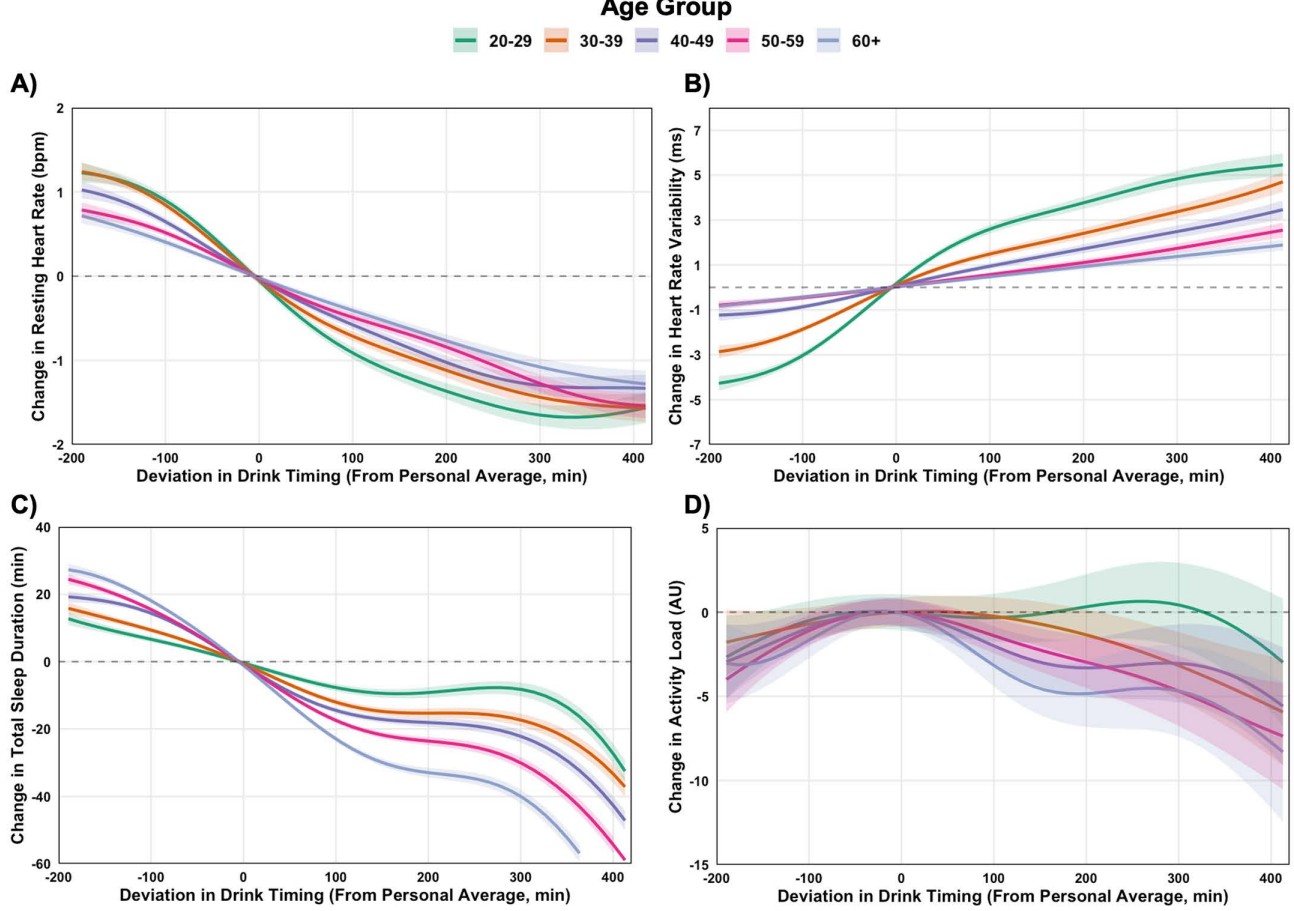

**Fig 5. Within-person associations between deviations in alcohol drink timing and physiological and behavioral outcomes, stratified by age group.** Generalized additive models estimated changes in resting heart rate (A), heart rate variability (B), sleep duration (C), and next-day activity (D) based on deviations from individuals' personal averages. Dose-response contrasts for drink timing are provided in S10 Table, with corresponding between-age group comparisons at different timing intervals in S11 Table.

Engaging in more physical activity than usual on the day of drinking was associated with amplified cardiovascular responses to alcohol (Fig 6D-6E). After five drinks more than usual, RHR was 0.7 bpm lower (99.9% CI: -0.9, -0.5) and HRV was 2.1 ms higher (99.9% CI: 1.6, 2.5; S13 Table) on low vs. high activity days (ES = 0.15 for RHR and ES = 0.16 for HRV). These effects were consistent across most alcohol consumption levels, with the greatest differences emerging at higher intakes. Effects on sleep were smaller in magnitude but followed a similar trend. After five drinks more than usual, engaging in lower than usual activity was associated with 4.3 min (99.9% CI: 1.7, 6.9) more sleep than those with higher than usual activity (ES = 0.06; Fig 6F).

Exploratory analyses examined whether self-reported hydration status moderated alcohol's acute effects (S6 Fig; S14 Table). Hydration modestly buffered cardiovascular disruptions. Following consumption of five drinks more than usual, RHR was 0.3 bpm lower (99.9% CI: 0.2, 0.5) and HRV was 0.9 ms higher (99.9% CI: 0.5, 1.3) on hydrated vs. non-hydrated nights (ES = 0.07 for RHR and ES = 0.07 for HRV). These effects were smaller at lower consumption levels but became more evident with heavier drinking. Hydration was also associated with slightly longer sleep duration after drinking. Effects on next-day physical activity were minimal.

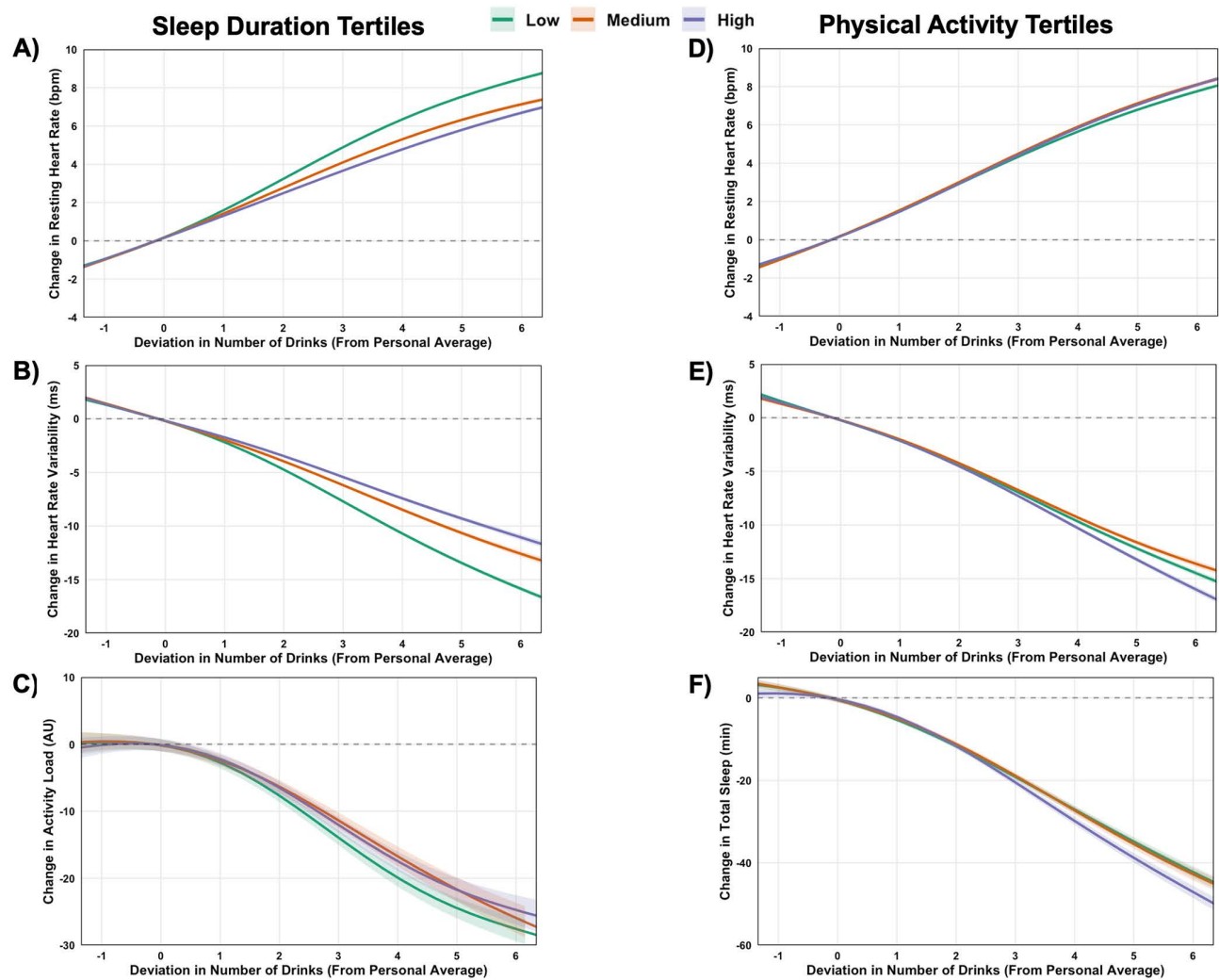

**Fig 6. Moderation of within-person associations between deviation in alcohol drink number and physiological and behavioral outcomes by sleep duration** (A-C) and physical activity tertiles (D-F). Generalized additive models estimated changes in resting heart rate (A&D), heart rate variability (B&E), physical activity (sleep moderation only; Panel C), and total sleep duration (exercise moderation only; Panel F) based on deviations from individuals' personal average number of drinks. For sleep models (A-C), nights after drinking were stratified into low (≤ −0.36 hours), moderate (> −0.36 to ≤ 0.49 hours), and high (> 0.49 hours) tertiles based on deviations from individuals' personal average sleep, with higher values reflecting more sleep than usual. For exercise models (D-F), exercise before drinking was stratified into low (≤ −41.81 AU), moderate (> −41.81 to ≤ 6.22 AU), and high (> 6.22 AU) tertiles based on deviations from individuals' personal average exercise volume, with higher values reflecting more exercise than usual. Contrasts between sleep durations at different drink quantities are presented in S12 Table, with contrasts between physical activity volume at different drink quantities provided in S13 Table.

## Discussion

In this real-world digital health study, acute alcohol consumption was associated with significant disturbances in cardio-vascular physiology, sleep, and next-day physical activity. Within persons, higher alcohol intake associated with dose-dependent increases in nocturnal RHR and reductions in HRV, indicating cardiac autonomic disruption. Alcohol use was also linked to shorter sleep and lower next-day physical activity. Effects were evident across sexes and age groups but were more pronounced in females and younger individuals. While speculative, these repeated transient physiological and behavioral insults may accumulate to adversely affect health over time.

Sex differences in cardiovascular, sleep, and physical activity responses to alcohol were consistently evident, with females exhibiting greater disturbances across most outcomes, even after adjusting for differences in body size and composition. These findings contrast with a prior real-world digital health study that reported no sex differences in cardiac autonomic responses to alcohol during the first three hours of sleep [10]. However, our results align with laboratory findings reporting greater HRV reductions among females following acute alcohol exposure [11], and observational data indicating a higher prevalence of heart disease among female, compared with male, drinkers [12]. This heightened sensitivity to alcohol in females may stem from lower first-pass metabolism of alcohol, largely attributable to lower gastric activity of glutathione-dependent formaldehyde dehydrogenase, together with a smaller distribution volume, which yields higher blood alcohol concentrations for a given dose [13]. Additional contributors may include lower lean body mass and potential hormonal influences [14,15]. Recognizing and quantifying these sex differences in a real-world setting supports sex-specific drinking guidance and highlights the value of targeted behavioral strategies to mitigate alcohol-related effects in females.

Participant age emerged as a clear moderator of alcohol's impact, with younger adults exhibiting more pronounced cardiovascular, sleep, and physical activity disruptions than older adults. This pattern is consistent with previous real-world observations from a Finnish cohort [10], and aligns with reports that hangover severity declines with age [16,17]. We speculate that age-related changes in the central nervous system, reflected by reduced cardiac autonomic responsiveness in the present study, may contribute to this apparent reduction of hangover severity. Additional contributors may include slower metabolic processing of alcohol with age [18], which may blunt acetaldehyde spikes [19], and age-related differences in alcoholic beverage choice [20]. Together, these factors may dampen acute cardiovascular, sleep, and physical activity responses to drinking in older adults.

Several modifiable health behaviors, including the timing of alcohol consumption, sleep duration after drinking, and physical activity on the day of drinking, emerged as meaningful moderators of alcohol's acute physiological and behavioral effects. These findings offer practical, evidence-based strategies to mitigate alcohol's physiological impact and its downstream effects on sleep and next-day physical activity. For example, consuming alcohol earlier in the evening, such as before rather than after dinner (approximately 2 hours earlier), was associated with improved nocturnal cardiac autonomic regulation, reflected by lower RHR and a higher HRV during sleep. Likewise, avoiding relatively high physical exertion (e.g., hard vs. light run) on drinking days and prioritizing longer sleep afterward were associated with more favorable overnight physiology. Intriguingly, we observed potential for a positive behavioral cycle whereby individuals who avoid excessive activity on the day of drinking tend to sleep longer that night (**Fig 6F**), and those who slept more exhibited higher activity levels the next-day (**Fig 6C**). This interconnection suggests that simple behavioral adjustments may both buffer alcohol's acute physiological effects and promote recovery patterns that support healthier behavioral routines.

Several limitations of this study should be acknowledged. First, alcohol intake was self-reported in a real-world setting, which introduces recall bias as well as uncertainty in dose and timing. Second, the observational design introduces the possibility there was confounding by co-occurring behaviors such as late-night eating or cannabis use [21,22]. Third, our sample comprises digital health users and may not represent the general population, which could limit generalizability. Relatedly, because all data came from a single, albeit validated wearable platform [23,24], findings may not extend to other devices. However, a recent longitudinal analysis reported comparable within-person associations between sleep and next-day physical activity in WHOOP and an external Fitbit cohort from the All of Us Research program [25], supporting cross-platform robustness. Fourth, many estimated effects were small in magnitude, consistent with acute physiological responses in real-world conditions. However, sequential dose-response contrasts reveal that small stepwise changes compound at higher intake levels. Fifth, we focused on acute physiological and behavioral consequences of alcohol using a within-person (via person-mean centering) framework. Accordingly, findings reflect short-term deviations from an individual's typical physiology and behavior rather than between-person differences or long-term health implications. An additional consideration relates to the conflict of interest. Several authors are employees of WHOOP, Inc., some of whom hold

stock options, and the research was funded by WHOOP, Inc. While the analyses were conducted independently by the research and data science teams and no commercial divisions were involved, it is important to acknowledge that demonstrating sensitivity of physiological measures to behavioral exposures, such as alcohol, could have favorable commercial ramifications for wearable devices and perhaps WHOOP in particular. Readers should therefore interpret these findings in this context, although the methodological transparency and prespecified analytical workflow help mitigate concerns regarding undue influence. Nonetheless, these limitations are counterbalanced by notable strengths. Our use of a validated wearable device with continuous data collection, combined with our within-subject design, enabled the precise and ecologically valid assessment of alcohol's impact while controlling for individual variability. The inclusion of next-day activity levels as an outcome adds a novel behavioral dimension, offering insight into alcohol's influence beyond sleep [26]. Finally, to our knowledge, this is the largest and most demographically rich investigation of alcohol's real-world physiological and behavioral effects to date, with stratification by both sex and age.

We provide large-scale, real-world evidence that alcohol consumption, even at moderate levels, disrupts overnight cardiac autonomic function, impairs sleep, and attenuates next-day physical activity. These effects were more pronounced in females and younger adults, but importantly, were moderated through simple behavioral strategies. At the population level, our findings support evolving public health recommendations that emphasize minimizing alcohol intake, while also offering simple yet effective guidance to mitigate health impacts when drinking does occur. Future research should focus on establishing causality through randomized controlled trials. Although there is compelling evidence demonstrating significant cardiovascular benefits associated with alcohol abstinence [27], further work is needed to develop scalable, evidence-based interventions that promote healthier drinking behaviors at both the individual and population levels.

## Materials and methods

### Study design and participants

We conducted a retrospective cohort study of de-identified data from adult users aged 20–100 years who activated a WHOOP Strap 4.0 (WHOOP Inc., Boston, MA) in 2023. All users consented for their anonymized data to be used for research purposes. To promote demographic balance, we performed stratified random sampling by biological sex (male/female) and age group (20–29, 30–39, 40–49, 50–59, ≥ 60 years), targeting 3,000 users per sex-by-age stratum.

Analyses used data from each participant's first 365 days after device activation to standardize follow up time, maximize data completeness, and minimize bias from variable retention during platform tenure. Eligibility required at least 84 nights of data within that 365-day window and at least one logged alcohol entry in the companion smartphone application. Following the initial SQL extraction, we removed individuals with fewer than 84 nights and those lacking alcohol journal responses during the window (**Fig 1**). Observation lengths ranged from 84 to 365 days (mean 244 days). The study was reviewed and approved by Salus IRB (Protocol #251121; Austin, TX) and is reported in accordance with STROBE guidelines.

### Wearable device and measures

The WHOOP strap continuously records heart rate via photoplethysmography and movement via three-axis accelerometer. RHR and HRV were derived from nocturnal sleep using a weighted average approach, prioritizing more stable periods of slow-wave sleep [28]. RHR was defined as the average number of heart beats per minute during the primary sleep episode. HRV was calculated as the root mean squared of successive beat-to-beat interval differences (RMSSD). To mitigate the influence of motion and signal artifacts, epochs identified as wake or as having low signal quality were excluded prior to aggregation. Remaining epochs were combined using a weighted average, with higher weights assigned to epochs with higher likelihood of slow-wave sleep and those occurring later in the sleep period. Sleep duration was defined as total sleep time. WHOOP (WHOOP, Inc. Boston, MA) has been validated against electrocardiography and polysomnography for heart rate (99% agreement) and 2-stage sleep categorization (86–89% agreement) [23,24].

Daily physical activity was calculated using the summated-heart-rate-zone score (SHRZS), a validated measure of internal training load developed by Edwards [29] and widely used across athletic and occupational populations [30–33]. Each participant's maximum heart rate was estimated using the non-linear formula 192–0·007*age^2 [34], manually set, or defined as their highest recorded heart rate. Physical activity was either detected automatically by the WHOOP analytics platform (~85% of cases) or logged manually by the participant to capture activities not automatically detected by the device. Importantly, regardless of detection method, activity intensity was derived from continuous heart rate data rather than manual input. Continuous heart rate data during each activity was used to calculate time spent in five zones defined as percentages of maximum heart rate: zone 1 (50–60%), zone 2 (60–70%), zone 3 (70–80%), zone 4 (80–90%), and zone 5 (90–100%). Minutes in each zone were multiplied by zone-specific weights (i.e., zone 1 = 1, zone 2 = 2, etc.) and summed to yield the SHRZS. This approach has demonstrated strong associations with session rating of perceived exertion and other internal and external load metrics across diverse populations [30–33].

## Alcohol consumption

Daily alcohol consumption was self-reported using the WHOOP smartphone application's journal. Each morning, users reported whether they consumed alcohol on the previous day (yes/no), how many drinks they consumed, and the time of their final drink. The journal also included a prompt about hydration, to which participants indicated whether they believed they hydrated adequately on the previous day (yes/no).

## Statistical analysis

Statistical analyses were performed in R (version 4.4.2). Outcome variables were assessed for outliers using the interquartile range (IQR) method, defined as values exceeding quartile 3 + 3xIQR or below quartile 1 - 3xIQR for each participant. This removed 0.02 to 2.1% of values per variable.

We used generalized additive models to examine associations between alcohol intake with physiological and behavioral responses. Outcomes included nocturnal RHR, HRV, sleep duration, and next-day physical activity. Independent variables were drink amount and timing of last drink relative to sleep onset. For the 5.1% of days missing drink amount we imputed the participant's median nonzero drink count. To separate within- and between-person effects, all variables were decomposed into daily deviations (within-person) and personal averages (between person) via person-mean centering.

Separate models were fit for each outcome of interest (RHR, HRV, sleep duration, and physical activity) with two independent variables (drink amount and drink timing). For each independent variable, we specified smooth terms interacting with either biological sex or age group in separate models, yielding 16 primary models (4 outcomes x 2 exposures x 2 moderators). Models focused on within-person effects, adjusting for between-person averages of outcomes and exposures. When age group was the moderator, it was included as a categorical covariate. When sex was the moderator, continuous age was modeled as a smooth term. In all models, the within-person component of the alternate exposure was included using a smoothing term, while the between-person component was entered as a linear covariate. Additional covariates included habitual alcohol use (proportion of drinking days; S1 Fig), body mass index (BMI), weekday vs. weekend, and season. To assess the robustness of our findings, models were re-estimated with height and weight instead of BMI. Results were unchanged, so BMI was retained in final models [35,36]. Because the analytic sample included nearly 21,000 participants with repeated observations, individual-level random intercepts could not be implemented at full scale. To evaluate whether omission of a random intercept affected inference, we re-estimated primary models in a random subsample of 5,000 participants including a subject-level random intercept. Effect estimates, P-values, and effect sizes were highly similar, supporting use of the fixed-effect specification in the full dataset. All variance inflation factors from simplified linear models were <2.

Model estimates were visualized with smoothing plots, and estimated marginal means were calculated at predefined within-person drink amounts (-1, 1, 3, 5) and timing offsets (-180, -60, 60, 180, 300, 420 min). Two contrast methods were used [37]: i) dose-response comparisons across exposures, and ii) between-group comparisons by sex or age group at

specific exposures. All pairwise comparisons used the multivariate t-distribution to control for family-wise error rate. Contrast estimates were standardized by dividing by the model residual standard deviation, yielding standardized effect sizes (ES). Sensitivity analyses were performed by repeating models using: i) absolute values for exposures and outcomes, and ii) stratifying participants by person-level median values of outcomes to examine whether underlying physiological state contributed to age or sex differences in alcohol responses.

Moderation analyses tested whether sleep duration, drinking-day physical activity, or reporting hydration modified alcohol's effects. Moderators were modeled separately to align with the preregistered analysis plan and to maintain interpretability. Exploratory analysis including both sleep and activity together yielded similar results, supporting the robustness of the primary findings. Sleep and activity were binned into tertiles of within-person deviation and modeled as categorical variables. Hydration was included as a binary variable and presented in supplemental materials. Separate models were fit to evaluate interactions between drink amount and each moderator, using the covariate structure described above. To conservatively account for 16 primary models, statistical significance was set at $\alpha = 0.001$, which is stricter than a Bonferroni correction of 0.003. Accordingly, 99.9% confidence intervals are reported.

## Supporting information

**S1 Table. Estimated alcohol drinking frequency differences in physiological and behavioral outcomes by number of drinks (within-person centered).**
(DOCX)

**S2 Table. Estimated differences in physiological and behavioral outcomes by number of drinks (within-person centered) by biological sex.**
(DOCX)

**S3 Table. Estimated biological sex differences in physiological and behavioral outcomes by number of drinks (within-person centered).**
(DOCX)

**S4 Table. Estimated biological sex differences in physiological and behavioral outcomes by number of drinks.**
(DOCX)

**S5 Table. Estimated differences in physiological and behavioral outcomes by time between last alcoholic drink and bedtime (within-person centered) by biological sex.**
(DOCX)

**S6 Table. Estimated biological sex differences in physiological and behavioral outcomes at varying times from last alcoholic drink to bedtime (within-person centered).**
(DOCX)

**S7 Table. Estimated differences in physiological and behavioral outcomes by number of drinks (within-person centered) by age group.**
(DOCX)

**S8 Table. Estimated age group differences in physiological and behavioral outcomes by number of drinks (within-person centered).**
(DOCX)

**S9 Table. Estimated age group differences in physiological and behavioral outcomes by number of drinks.**
(DOCX)

**S10 Table. Estimated differences in physiological and behavioral outcomes by time between last alcoholic drink and bedtime (within-person centered) by age group.**
(DOCX)

**S11 Table. Estimated age group differences in physiological and behavioral outcomes at varying times from last alcoholic drink to bedtime (within-person centered).**
(DOCX)

**S12 Table. Estimated sleep tertile differences in physiological and behavioral outcomes by number of drinks (within-person centered).**
(DOCX)

**S13 Table. Estimated activity tertile differences in physiological and behavioral outcomes by number of drinks (within-person centered).**
(DOCX)

**S14 Table. Estimated hydration differences in physiological and behavioral outcomes by number of drinks (within-person centered).**
(DOCX)

**S1 Fig. Moderation of within-person associations between deviation in alcoholic drink number and physiological and behavioral outcomes by drinking frequency.**
(DOCX)

**S2 Fig. Sensitivity analysis examining whether biological sex-related differences in physiological and behavioral responses to alcohol vary by absolute drink amount.**
(DOCX)

**S3 Fig. Sensitivity analysis examining whether biological sex-related differences in physiological and behavioral responses to alcohol vary by baseline physiology.**
(DOCX)

**S4 Fig. Sensitivity analysis examining whether age group-related differences in physiological and behavioral responses to alcohol vary by absolute drink amount.**
(DOCX)

**S5 Fig. Sensitivity analysis examining whether age group-related differences in physiological and behavioral responses to alcohol vary by baseline physiology.**
(DOCX)

**S6 Fig. Moderation of within-person associations between deviation in alcoholic drink number and physiological and behavioral outcomes by hydration status.**
(DOCX)

## Acknowledgments

The authors wish to acknowledge the technical assistance of the WHOOP Data Science and Data Engineering teams for their contributions to feature development and data infrastructure. We also thank the broader WHOOP employee community for their continued support of the company's research initiatives.

## Author contributions

**Conceptualization:** Gregory James Grosicki, Austin T. Robinson, Michael J. Joyner, Jason R. Carter, William von Hippel, David M. Presby, Finnbarr Fielding, Jeremy A. Bigalke, Jeongeun Kim, Christopher Chapman, Kristen E. Holmes.

**Data curation:** Gregory James Grosicki, Jeongeun Kim.

**Formal analysis:** Gregory James Grosicki, Jeongeun Kim.

**Investigation:** Gregory James Grosicki.

**Methodology:** Gregory James Grosicki.

**Visualization:** Gregory James Grosicki.

**Writing – original draft:** Gregory James Grosicki, Austin T. Robinson, Michael J. Joyner, Jason R. Carter, William von Hippel, David M. Presby, Finnbarr Fielding, Jeremy A. Bigalke, Jeongeun Kim, Christopher Chapman, Kristen E. Holmes.

**Writing – review & editing:** Gregory James Grosicki, Austin T. Robinson, Michael J. Joyner, Jason R. Carter, William von Hippel, David M. Presby, Finnbarr Fielding, Jeremy A. Bigalke, Jeongeun Kim, Christopher Chapman, Kristen E. Holmes.

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
