## [Decision Letter · Decision Letter 0]

18 Dec 2025

Response to Reviewers
Revised Manuscript with Track Changes
Manuscript
**Journal Requirements:**
**Additional Editor Comments (if provided):**

However, the reviewers raised several substantive methodological and reporting concerns that require careful attention before the manuscript can be recommended for publication. First, the conflict of interest warrants more substantive discussion. Seven of the eleven authors, including the first and senior authors, are employees of WHOOP, Inc., the manufacturer of the device used in this study. While disclosed, the statement that "WHOOP has no stake in the outcome of this research" inadequately acknowledges that research demonstrating a commercial device's sensitivity to lifestyle factors inherently promotes the device's utility and market value. The limitations section should more transparently address how this conflict was managed and its potential implications for data interpretation.

Second, the reviewers identified critical statistical issues that must be addressed. Given the very large sample size, statistical significance testing yields extremely small p-values for virtually all comparisons, including those in the fourteen supplementary tables. The manuscript needs to shift emphasis from statistical significance to effect sizes and clinical or practical significance, providing readers with clear guidance on which observed differences are likely to be meaningful in real-world contexts. More problematically, the study reports results from 16 primary models plus extensive supplementary analyses without applying appropriate corrections for multiple comparisons. While the authors set alpha at 0.005 rather than 0.05, this does not address the family-wise error rate inflation from testing multiple hypotheses. The reviewers questioned why Bonferroni or similar corrections were not applied across the model family, and why all hypothesis tests in Tables S1-S14 are not factored into a comprehensive multiple comparison correction. This is a fundamental statistical concern that undermines confidence in which findings represent true effects versus Type I errors.

Third, several methodological choices require better justification or revision. The timing of sampling (after device activation but before checking eligibility criteria) is inefficient and should be clarified. The decision to model sleep and physical activity moderators separately rather than jointly limits understanding of their potential interactions. The choice of person-mean centring, while reasonable, has implications for interpretation that should be acknowledged in the limitations. The composite physical activity score, while previously used by this research group, would benefit from additional supporting references to establish its broader validity.

Fourth, important measurement and data quality issues need clarification. The manuscript does not describe how artifacts were handled in the RMSSD calculation, despite HRV's known sensitivity to measurement artifacts. The protocol for reconciling manual physical activity entries with automated detection requires explanation. The limitations should acknowledge recall bias inherent in manual alcohol consumption logging. Table 1 reports numerous statistically significant comparisons despite very large standard deviations, which appears inconsistent and likely reflects the enormous sample size rather than meaningful differences; this reinforces the need to emphasize effect sizes throughout.

**Reviewers' Comments:**

**Comments to the Author**

1. Does this manuscript meet PLOS Digital Health’s publication criteria?

Reviewer #1: Yes

Reviewer #2: Yes

2. Has the statistical analysis been performed appropriately and rigorously?

Reviewer #1: Yes

Reviewer #2: No

3. Have the authors made all data underlying the findings in their manuscript fully available (please refer to the Data Availability Statement at the start of the manuscript PDF file)?

Reviewer #1: No

Reviewer #2: Yes

4. Is the manuscript presented in an intelligible fashion and written in standard English?

Reviewer #1: Yes

Reviewer #2: Yes

Reviewer #1: The manuscript presents a robust analysis of a very large, real-world dataset from a commercial wearable device to quantify the acute effects of alcohol use on nocturnal cardiovascular measures (RHR, HRV), sleep, and next-day activity levels. A key strength is the within-person design, analyzing over 5 million person-days from nearly 21,000 participants to isolate the effects of alcohol relative to a user's own baseline. The study demonstrates that alcohol increases RHR, decreases HRV, and modestly reduces sleep and activity, with a novel finding that these effects are more pronounced in females and younger adults. The investigation into some of the “mitigating” factors (e.g., drink timing, sleep duration) provides some practical insights.

Major Comment

The study's funding and the employment of 7 of 11 authors (including the first and senior authors) by the device manufacturer (WHOOP, Inc.) represent a significant conflict of interest. While disclosed, this COI could be more substantially addressed in the limitations. The current disclosure states "WHOOP has no stake in the outcome of this research", which is debatable, as findings that highlight the device's sensitivity to lifestyle factors (like alcohol) inherently promote the device's utility and value.

Minor Comments

Given the very large N, nearly all tests are statistically significant. Please ensure that the text (stats, and especially in the Discussion) focuses on the magnitude and clinical significance of these effects.

Please consider placing the METHODS section after the Introduction, as is typical of most major journals.

Reviewer #2: Line 62 - I suggest using ":" instead of ". For example,"

Lines 87-92 - These should go in the Methods section, that should be after the Introduction

Line 229 - Add to limitations the recall bias of inputing alcohol consumption manually

Line 236 - Add to limitations the effect of the choice of person-mean centering

Line 258 - Why wasn't the sampling performed after checking for eligibility criteria?

Line 266 - STROBE guidelines suggest to indicate the study design in the title or abstract

Line 272 - How were artifacts handled? RMSSD is sensitive to artifacts.

Line 278 - How was manual input for physical activity handled with when there was contextual automated detection?

Line 283 - Are there other references about the usage of a composite score?

Line 302 - Why weren't both the moderators included in the model, but modeled separately?

Line 326 - Why weren't the Bonferroni correction or other methods used, which would decrease the p-value (being 16 models)?

Table 1 - Are the numbers after ± standard deviations? How can most of the comparisons be statistically significant when the standard deviations are so high?

Tables S1-S14: All the p-values reported are hypothesis tested. They should be factored into the correction for multiple comparisons

**Do you want your identity to be public for this peer review?** For information about this choice, including consent withdrawal, please see our Privacy Policy

Reviewer #1: No

Reviewer #2: No

**Figure resubmission:**

**Reproducibility:** To enhance the reproducibility of your results, we recommend that authors of applicable studies deposit laboratory protocols in protocols.io, where a protocol can be assigned its own identifier (DOI) such that it can be cited independently in the future. Additionally, PLOS ONE offers an option to publish peer-reviewed clinical study protocols. Read more information on sharing protocols at https://plos.org/protocols?utm_medium=editorial-email&utm_source=authorletters&utm_campaign=protocols

---

## [Decision Letter · Decision Letter 1]

19 Feb 2026

Real-World Effects of Alcohol on Heart Rate, Sleep, and Physical Activity by Age and Sex

PDIG-D-25-00847R1

Dear Gregory James Grosicki,

We are pleased to inform you that your manuscript 'Real-World Effects of Alcohol on Heart Rate, Sleep, and Physical Activity by Age and Sex' has been provisionally accepted for publication in PLOS Digital Health.

Best regards,

Cleva Villanueva, M.D., Ph.D.

Academic Editor

PLOS Digital Health

**Additional Editor Comments (if provided):**

he reviewers who evaluated the first version of the manuscript agreed that the authors appropriately addressed all their comments and revised the manuscript accordingly. After assessing both the original and the revised versions, and considering the reviewers’ feedback, the article is now suitable for publication in PLOS Digital Health

**Reviewer Comments (if any, and for reference):**

Reviewer's Responses to Questions

**Comments to the Author**

Reviewer #1: All comments have been addressed

Reviewer #2: All comments have been addressed

publication criteria?

Reviewer #1: Yes

Reviewer #2: Yes

3. Has the statistical analysis been performed appropriately and rigorously?

Reviewer #1: Yes

Reviewer #2: Yes

4. Have the authors made all data underlying the findings in their manuscript fully available (please refer to the Data Availability Statement at the start of the manuscript PDF file)?

Reviewer #1: Yes

Reviewer #2: No

5. Is the manuscript presented in an intelligible fashion and written in standard English?

Reviewer #1: Yes

Reviewer #2: Yes

Reviewer #1: I have no further recommendations.

Reviewer #2: The revised manuscript satisfactorily addresses all revision requests.

**Do you want your identity to be public for this peer review?** For information about this choice, including consent withdrawal, please see our Privacy Policy

Reviewer #1: No

Reviewer #2: No
